# Osteoarthritis in older adults: A global health challenge and the role of high BMI in shaping disease trends

Yan Gao[1], Hailong Yu[2], Wenfeng Han[2], Ning Wang[2], Bin Zheng[2], Aoxiang Yang[2], Yu Wang[2]*

**1** Department of Disease Prevention and Control, General Hospital of Northern Theater Command, Shenyang, China, **2** Department of Orthopedics, General Hospital of Northern Theater Command, Shenyang, China

* wangyu110016@163.com

## Abstract

### Background

This study aimed to estimate the global, regional, and national Osteoarthritis (OA) burden and trends in older adults from 1990 to 2021, with a particular focus on the contribution of high body mass index (BMI) to the OA burden.

### Methods

Using 2021 Global Burden of Disease Study data, we estimated the incidence, DALYs, and trends of OA in older adults across 204 countries and territories from 1990 to 2021. Spearman's correlation analysis was applied to investigate relationships between age-standardized rates and the sociodemographic index (SDI). We quantified the BMI-attributable contribution to the OA burden. The Bayesian Age-Period-Cohort (BAPC) model was applied to project the global OA trend in older adults up to 2040.

### Results

From 1990 to 2021, global OA cases in older adults increased by 148.65%. Age-standardized incidence and DALY rates increased annually by 0.08% and 0.31%, respectively. East and Southeast Asia exhibited the fastest increase in age-standardized DALY rates. The OA burden peaked in the 65–69-year age group, with women compared with men experiencing a higher burden. Knee OA imposed the heaviest burden, followed by hand OA, with regional variations across four anatomical sites. The OA burden, including specific sites, positively correlated with the SDI. The attributable proportion of OA burden due to high BMI increased by 21.39%, with the knee OA burden from high BMI exceeding the hip OA burden. The BAPC model predicted that OA burden will continue to rise in the future.

**Data availability statement:** All relevant data are within the paper and its Supporting information files.

**Funding:** This work was supported by the Liaoning Provincial Science and Technology Plan Joint Program (Applied Basic Research Project) (Grant No. 2023JH2/101700124) and the Liaoning Provincial Science and Technology Plan Joint Program (Technology Research and Development Project) (Grant No. 2024JH2/102600270). The corresponding author (Yu Wang) is the recipient of both grants. The funders had no role in study design, data collection and analysis, decision to publish, or preparation of the manuscript.

**Competing interests:** The authors have declared that no competing interests exist.

## Conclusions

OA in older adults is a major global challenge, with knee OA being the most burdensome. High BMI, a modifiable risk factor, contributes significantly to OA, particularly in the knees. To reduce the OA burden, public health strategies should prioritize obesity prevention through weight management and awareness campaigns for older adults.

## Introduction

Osteoarthritis (OA) is one of the most common types of arthritis [1], often considered an age-related disease and a leading cause of chronic pain and long-term disability among adults worldwide [2–4]. According to previous findings from the Global Burden of Disease (GBD) Study, the age-standardized annual incidence rate of OA increased by 8.2% from 1990 to 2017 [5]. By 2020, approximately 595 million people worldwide were affected by OA, making it the seventh leading cause of years lived with disability (YLD) among adults aged ≥70 years [6], significantly impacting the health and quality of life of older adults.

Ageing is one of the primary factors influencing OA onset. As joints age, their ability to resist damage gradually diminishes [7]. The characteristics of OA vary across sex and anatomical sites. Studies have shown that OA is significantly more prevalent among female individuals than among male individuals [8], with this sex disparity being most pronounced in knee OA [9]. OA prevalence also differs by the affected joint, with the knee being the most commonly affected owing to its weight-bearing nature, followed by the hand and hip joints [10]. A high body mass index (BMI) is a major risk factor for knee OA [11], whereas hand OA is associated with systemic inflammation [12]. These findings highlight the importance of conducting burden studies focusing on different anatomical sites of OA and specific populations.

Previous studies have examined the disease burden of OA at global, regional, and national levels; however, research findings on incidence rates and disability-adjusted life years (DALYs) remain insufficient [13,14]. Crucially, systematic analyses specifically focusing on older adults—the population at highest risk—remain relatively limited. Most prior studies have not utilized the latest GBD 2021 data to analyze trends over the past three decades, nor have they adequately quantified the burden attributable to high BMI or projected future trends specific to the geriatric population. This gap poses challenges to comprehensively understanding the characteristics and changing patterns of OA burden among older adults.

To address these gaps, this study utilizes the most recent data from the GBD 2021 study. We selected the period from 1990 to 2021 to capture the most comprehensive long-term historical trends available and extended our projections to 2040. This projection horizon was chosen to provide a roughly 20-year window for policymakers to anticipate healthcare demands associated with global population ageing and to align with long-term public health planning goals. The specific objectives of this study were as follows: (1) evaluation of global, regional, and national patterns and long-term trends in the incidence and DALYs of OA among older adults from 1990 to

2021, stratified by sex, age group, and anatomical site; (2) quantification of the contribution of high BMI to the OA burden in older adults, providing evidence for this key modifiable risk factor; and (3) projection of the future disease burden of OA in older adults through 2040 using the BAPC model, thereby offering data-driven insights to support public health decision-making and resource allocation.

## Methods

### Study population and data collection

In the GBD 2021 study, the estimation process utilized data from a total of 100,983 sources, with 19,189 new data sources added in 2021. This extensive dataset provides the latest estimates on the epidemiology of 371 diseases and injuries across 21 GBD regions and 204 countries and territories, covering the period from 1990 to 2021. It includes critical data from surveys, disease registries, case reports, and hospital records. Key datasets from the Global Health Data Exchange (GHDx), such as the Demographic and Health Surveys (DHS), have played a vital role in capturing diverse health outcomes across various regions and populations. Among these sources, 75,459 specifically address nonfatal conditions, encompassing key epidemiological indicators such as incidence and DALYs [15,16]. All data came from reputable public databases and underwent strict screening to ensure quality. Furthermore, the authors of the GBD study have committed to updating the data annually to ensure its accuracy [17].

The GBD 2021 collaborators modeled estimates of incidence and mortality for individuals of all ages and sexes. The GBD team developed the DisMod-MR software (version 2.1), a Bayesian meta-regression tool used for estimating incidence by analyzing a cascade process. Prior to modeling, data points and biases were adjusted in the following ways: (1) data that were not broken down by age and sex was disaggregated; (2) a Bayesian, regularized, and trimmed meta-regression model was applied to directly compare study designs and case definitions. For information on bias correction and other modifications applied to each specific disease, refer to the summary report of GBD 2021 [18]. DALYs, a composite measure of overall health loss, were calculated by summing years of life lost (YLLs) and years lived with disability (YLDs). YLLs were calculated by multiplying the number of deaths associated with specific causes by the remaining life expectancy at the time of death, based on a standard life expectancy [19].

The 2021 GBD Study separately estimated OA in the hip, knee, hand, and other anatomical sites. Although spinal OA is also common, any symptoms and disabilities associated with the cervical, thoracic, and/or lumbar spine are expected to be included in the estimates for low back and neck pain. In the 2021 GBD study, the reference case definition for OA is symptomatic OA confirmed by radiography, with a Kellgren–Lawrence grade of 2–4 [20]. The International Classification of Diseases, 10th Revision (ICD-10) codes for OA are M16–M19. The complete ICD-10 codes can be referenced at 'https://ghdx.healthdata.org/record/ihme-data/gbd-2021-cause-icd-code-mappings'.

Data included for OA in the 2021 GBD study consist of state-level insurance claims data from the USA for 2000 and 2010–2016, as well as insurance claims data from Taiwan for 2016 [6]. Relevant data sources can be obtained through the 2021 GBD study data input sources tool (https://ghdx.healthdata.org/gbd-2021/sources) on the Institute for Health Metrics and Evaluation website. OA incidence was assumed to be zero before the age of 30 years. In the models for hip, knee, and other types of OA, the age-standardized summary exposure value scale for OA (a normalized measure of disease-related risk) and BMI were included as covariates for prevalence. Detailed information about the OA disease model can be found in the 2021 GBD study methods appendix (https://www.healthdata.org/gbd/methods-appendices-2021/osteoarthritis).

In this study, we extracted data from the Global Burden of Disease (GBD) Study 2021 results tool (https://vizhub.healthdata.org/gbd-results/). Annual data on the absolute numbers and rates of incidence and DALYs for OA and its four anatomical sites (knee, hip, hand, and other OA) were systematically retrieved from 1990 to 2021. The dataset encompasses 204 countries and territories, which are further aggregated into 21 GBD regions based on epidemiological similarity and geographic proximity. Data were stratified by sex and seven age groups (65–69, 70–74, 75–79, 80–84, 85–89, 90–94, and

≥95 years). We utilized the 95% uncertainty intervals (UIs) provided by the GBD 2021 study for all estimates. In the estimation process for OA, uncertainty is propagated through each step, with UIs representing the 2.5th and 97.5th percentiles of the 1000-sample distribution. We selected the population aged ≥65 years as the study's focus population based on the following scientific rationale: previous studies have indicated that age is one of the most prominent risk factors for OA [21]. Most individuals aged ≥65 years are affected by OA [22,23], with one in three being diagnosed with the condition [24]. Furthermore, patients with OA aged ≥65 years are more likely to require surgeries (such as joint replacement), medication, and physical therapy [25].

The sociodemographic index (SDI) is a composite indicator, with the SDI in the 2021 GBD study ranging from 0 (lowest income and years of education, highest fertility rate) to 100 (highest income and years of education, lowest fertility rate) [26]. High BMI is identified as a risk factor for arthritis, including hip and knee OA, and is defined as a BMI ≥ 25 kg/m² [6]. Detailed data and methods regarding high BMI can be found elsewhere [27].

## Statistical analysis

First, we compared the age-standardized incidence rate (ASIR; per 100,000 population) and the age-standardized DALY rate (ASDR; per 100,000 population) of OA and its four anatomical sites (hip, knee, hand, and other sites) in older adults across different regions, countries, sexes, and age groups globally. The formula used for calculating the age-standardized rate (ASR) of OA, expressed per 100,000 population, was as follows:

$$\frac{\sum_{i=1}^{A} \alpha_i W_i}{\sum_{i=1}^{A} W_i}$$

($\alpha_i$: the age-specific rate in $i$th the age group; $W$: the number of people in the corresponding $i$th age group among the GBD standard population; $A$: the number of age groups).

This study used the estimated annual percentage change (EAPC) to evaluate the average trend in the ASR of OA from 1990 to 2021 [28]. If the EAPC is positive and its 95% confidence interval (CI) excludes zero, the ASR shows a significant increasing trend; if the EAPC is negative and its 95% CI excludes zero, it indicates a significant decreasing trend; and if the 95% CI includes zero, the change is not significant.

We used the local regression smoothing model to fit the correlation between OA burden (including its four anatomical sites) and the SDI in older adults globally and in 21 regions from 1990 to 2021. Additionally, Spearman's correlation analysis was used to calculate the correlations between the ASIR, ASDR, EAPC, and SDI in 2021, determining the correlation coefficient (r) and P-value. A P-value <0.05 was regarded as statistically significant.

According to GBD 2021, high BMI was the sole risk factor analyzed for its causal association with the risk of total OA, knee OA, and hip OA. The framework utilized the population attributable fraction (PAF) methodology to quantify high BMI-related OA burden and reported the corresponding ASDRs. To assess the proportional contribution of high BMI to this burden, we calculated the proportion of ASDRs attributable to high BMI relative to the overall ASDRs of total OA, knee OA, and hip OA in older adults globally, across 21 regions, and in 204 countries and territories. Additionally, we analyzed these attributable proportions stratified by age groups at the global and regional levels.

In addition, a BAPC model was used to project global trends in OA burden among older adults up to 2040. The model decomposes temporal variation into age, period, and cohort effects, which were assumed to evolve smoothly over time and were modeled using second-order random walk priors, as implemented in the BAPC framework [29]. Posterior inference was performed using Integrated Nested Laplace Approximation (INLA), which provides deterministic approximations to marginal posterior distributions. All model specifications followed the default settings of the "BAPC" R package. Population projections were obtained from the IHME database and incorporated to account for future demographic changes. Projections were intended to describe long-term trends under current epidemiological patterns rather than to infer causal

effects. All statistical analyses and graphical visualizations were performed using R software (version 4.2.3; R Foundation for Statistical Computing, Vienna, Austria).

### Ethics statement

For GBD studies, the Institutional Review Board of the University of Washington reviewed and approved a waiver for informed consent (https://www.healthdata.org/research-analysis/gbd).

## Results

### Global OA burden in older adults

Globally, OA incident cases in older adults increased by 148.65%, from 4.48 million in 1990 to 11.14 million in 2021. By 2021, the ASIR of OA in this age group had reached 1,446.09 per 100,000 population. Meanwhile, DALYs due to OA increased by 157.16%, from 3.84 million to 9.87 million during the same period. By 2021, the ASDR of OA in this age group had reached 1,281.99 per 100,000 population (Table 1). In 2021, among older adults, female individuals had a significantly higher ASR of OA than male individuals.

From 1990 to 2021, the global ASIR and ASDR of OA in older adults showed an increasing trend. During this period, the increase in ASIR among female individuals was slightly lower than that among male individuals; however, the increase in ASDR among female individuals was higher than that among male individuals (Table 1).

Based on OA distribution across four anatomical sites, the incidence and DALYs of OA in older adults globally in 1990 and 2021 were ranked as follows: knee OA, hand OA, OA in other sites, and hip OA. In 2021, the number of incident cases and their corresponding proportions were 7.48 million (67.1%), 2.14 million (19.3%), 0.98 million (8.8%), and 0.54 million (4.8%), respectively. The corresponding DALYs and their proportions were 5.20 million (52.7%), 3.07 million (31.1%), 1.02 million (10.4%), and 0.58 million (5.8%) (Fig 1). Between 1990 and 2021, hand OA showed the most significant increase in ASR among the anatomical sites, with its ASIR increasing from 246.13 to 278.37 per 100,000 population, while its ASDR increased from 350.07 to 398.84 per 100,000 population (S1 Table in S1 File).

### Regional OA burden in older adults

Across the 21 GBD study regions, from 1990 to 2021, the OA burden in older adults showed varying degrees of increase in all regions except for high-income Asia-Pacific, Western Europe, and high-income North America. In 2021, East Asia, South Asia, and Western Europe ranked highest in OA incidence cases and DALYs. Notably, the region with the highest ASR of OA was high-income Asia-Pacific (S1 Fig in S1 File).

Regarding OA distribution across the four anatomical sites, in both 1990 and 2021, knee and hand OA accounted for most incident cases and DALYs in older adults across the 21 GBD study regions (Fig 1). Among these, East and South Asia had the highest incidence of knee OA cases and DALYs in 2021. Specifically, the number of new knee OA cases in East and South Asia was 1.85 million (66.4%) and 1.15 million (68.0%), respectively, with corresponding DALY burdens of 1.43 million (59.9%) and 0.69 million (52.5%) (S1 Table in S1 File; Fig 1).

From 1990 to 2021, hand OA in East Asia among older adults showed the fastest ASR increase, with an EAPC of 0.82 (95% CI: 0.65–1.00) for ASIR and 1.52 (95% CI: 1.34–1.71) for ASDR. The largest ASR increase for knee OA was observed in Southeast Asia, with an EAPC of 0.35 (95% CI: 0.33–0.36) for ASIR and 0.46 (95% CI: 0.44–0.48) for ASDR (S1 Table in S1 File).

### National OA burden in older adults

In 2021, China, India, and the USA ranked as the top three countries for OA incident cases and DALYs in older adults. In the same year, Brunei Darussalam had the highest ASIR, at 1,896.14 per 100,000, whereas South Korea recorded

**Table 1. Incidence and disability-adjusted life-years of osteoarthritis and their estimated annual percentage changes among older adults globally and in 21 GBD regions.**

| Characteristics | Incidence | | | | | DALYs | | | | |
|---|---|---|---|---|---|---|---|---|---|---|
| | Number of cases, 1990 | ASR per 100,000 population, 1990 | Number of cases, 2021 | ASR per 100,000 population, 2021 | EAPC, 1990–2021 | Number of cases, 1990 | ASR per 100,000 population, 1990 | Number of cases, 2021 | ASR per 100,000 population, 2021 | EAPC, 1990–2021 |
| Global | 4,480,350.91 (3,419,778.82–5,696,093.44) | 1,375.96 (1,049.13–1,750.76) | 11,137,926.45 (8,470,935.51–14,185,354.52) | 1,446.09 (1,099.08–1,842.71) | 0.08 (0.03–0.12) | 3,840,423.27 (1,867,589.89–7,823,864.01) | 1,179.43 (572.73–2,404.47) | 9,874,039.44 (4,793,549.21–20,091,607.09) | 1,281.99 (621.81–2,609.73) | 0.31 (0.28–0.34) |
| **Sex** | | | | | | | | | | |
| Male | 1,662,599.43 (1,268,936.70–2,117,863.72) | 1,187.15 (904.49–1,514.27) | 4,417,446.84 (3,372,897.57–5,622,112.50) | 1,264.81 (964.71–1611.07) | 0.10 (0.04–0.15) | 1,321,246.93 (642,052.84–2,681,353.24) | 943.42 (457.33–1,916.87) | 3,636,049.24 (1,764,418.24–7,375,412.87) | 1,041.08 (504.45–2,113.27) | 0.29 (0.26–0.33) |
| Female | 2,817,751.48 (2,156,295.24–3,584,272.38) | 1,518.45 (1,160.45–1,933.52) | 6,720,479.62 (5,117,435.80–8,577,630.45) | 1596.49 (1214.62–2039.03) | 0.09 (0.06–0.13) | 2,519,176.34 (1,224,617.26–5,140,488.75) | 1,357.56 (658.76–2,772.55) | 6,237,990.20 (3,028,002.83–12,702,176.86) | 1,481.87 (718.51–3,019.13) | 0.36 (0.32–0.40) |
| **Site of osteoarthritis** | | | | | | | | | | |
| Hand | 801,425.19 (397,196.30–415,309.20) | 246.13 (121.6–435.37) | 2,144,066.6 (1,066,018.85–3,775,070.13) | 278.37 (138.14–490.63) | 0.29 (0.22–0.35) | 1,139,897.4 (524,804.01–2,348,554.24) | 350.07 (160.74–722.19) | 3,071,942.05 (1,418,456.93–6,339,314.89) | 398.84 (183.86–823.70) | 0.49 (0.41–0.58) |
| Hip | 212,792.59 (113,611.50–363,518.20) | 65.35 (34.69–112.00) | 540,023.21 (283,400.49–933,698.04) | 70.11 (36.66–121.47) | 0.16 (0.12–0.19) | 231,709.29 (110,148.00–468,279.12) | 71.16 (33.63–144.23) | 576,557.75 (272,978.39–1,165,460.74) | 74.86 (35.31–151.59) | 0.24 (0.21–0.27) |
| Knee | 3,054,148.64 (2,200,172.57–4,146,258.76) | 937.96 (674.8–1,274.58) | 7,478,349.2 (5,369,036.97–10,147,887.85) | 970.95 (696.5–1,318.36) | 0.02 ($-0.02$ to 0.07) | 2,060,496.14 (995,729.28–4,121,883.72) | 632.8 (305.2–1267.09) | 5,200,673.76 (2,515,827.52–10,394,733.38) | 675.23 (326.24–1,350.42) | 0.23 (0.21–0.25) |
| Other | 411,984.49 (238,905.70–558,049.51) | 126.52 (73.08–171.83) | 975,487.36 (566,006.35–1,319,345.37) | 126.65 (73.3–171.59) | 0.00 (0.00–0.00) | 408,320.40 (189,685.34–868,619.58) | 125.4 (57.99–267.32) | 1,024,865.88 (478,103.81–2,176,080.16) | 133.06 (61.90–282.91) | 0.22 (0.20–0.23) |
| **GBD regions** | | | | | | | | | | |
| East Asia | 1,649,409.69 (1,151,899.61–2,267,798.30) | 1,226.89 (917.34–1,585.08) | 5,558,154.04 (3,877,777.58–7,653,243.95) | 1,371.11 (1,031.07–1,760.55) | 0.24 (0.13–0.36) | 1,332,162.60 (632,952.64–2,693,328.16) | 990.91 (471.02–2,007.18) | 4,763,608.45 (2,265,457.25–9,664,672.40) | 1,175.11 (560.75–2,385.38) | 0.66 (0.56–0.75) |
| South Asia | 1,044,237.89 (744,604.69–1,415,905.11) | 1,293.53 (983.74–1,651.81) | 3,380,380.57 (2,404,235.80–4,592,721.53) | 1,413.17 (1,076.24–1,804.76) | 0.28 (0.26–0.3) | 1,506,587.49 (721,260.78–3,053,608.15) | 919.64 (444.38–1,865.11) | 5,323,631.89 (2,552,437.25–10,797,591.61) | 1,100.55 (532.73–2,229.75) | 0.59 (0.56–0.62) |
| Southeast Asia | 391,816.95 (273,055.43–539,961.93) | 1,058.09 (796.17–1,359.12) | 1,212,683.43 (846,080.69–1,671,547.81) | 1,185.23 (895.44–1,519.43) | 0.39 (0.38–0.40) | 301,346.76 (143,829.53–611,851.69) | 813.78 (390.97–1,653.91) | 1,004,189.81 (479,456.35–2,043,201.16) | 981.45 (474.76–1,994.60) | 0.66 (0.64–0.67) |
| Central Asia | 86,233.00 (60,247.58–118,667.71) | 1,234.11 (935.25–1,579.23) | 156,986.02 (110,653.13–214,701.45) | 1,303.75 (996.21–1,658.4) | 0.11 (0.07–0.14) | 84,004.68 (39,998.45–171,026.05) | 1202.22 (577.24–2,450.12) | 160,077.96 (76,276.73–324,918.26) | 1,329.43 (639.34–2,685.29) | 0.45 (0.38–0.53) |

*(Continued)*

Table 1. (Continued)

| Characteristics | Incidence | | | | | DALYs | | | | |
|---|---|---|---|---|---|---|---|---|---|---|
| | Number of cases, 1990 | ASR per 100,000 population, 1990 | Number of cases, 2021 | ASR per 100,000 population, 2021 | EAPC, 1990–2021 | Number of cases, 1990 | ASR per 100,000 population, 1990 | Number of cases, 2021 | ASR per 100,000 population, 2021 | EAPC, 1990–2021 |
| High-income Asia-Pacific | 634,716.98 (457,992.37–855,334.05) | 1,826.7 (1,404.88–2,326.83) | 1,578,172.64 (1,127,242.47–2,137,138.75) | 1,710.27 (1,303.5–2,182.72) | -0.33 (-0.38 to -0.28) | 565,122.35 (271,673.95–1,148,503.61) | 1,626.41 (785.82–3,320.68) | 1,627,523.95 (780,641.25–3,311,579.63) | 1,763.76 (850.75–3,598.05) | 0.52 (0.31–0.72) |
| Oceania | 4,518.90 (3,208.57–6,177.30) | 1,149.73 (842.44–1,531.19) | 12,040.30 (8,476.60–16,459.48) | 1,231.59 (907.11–1,610.07) | 0.21 (0.2–0.22) | 3,588.71 (1,710.10–7,316.85) | 913.07 (412.08–1,925.06) | 10,021.55 (4,784.85–20,333.76) | 1,025.09 (475.07–2,118.71) | 0.36 (0.32–0.39) |
| Australasia | 65,599.83 (46,885.23–88,901.05) | 1,470.05 (1108.83–1,886.1) | 161,210.97 (114,806.42–218,468.41) | 1,551.23 (1,176.42–1,993.03) | 0.09 (0.04–0.13) | 61,224.94 (29,419.15–124,249.61) | 1,372.01 (657.72–2,801.2) | 157,436.30 (76,174.17–320,212.26) | 1,514.91 (737.46–3,094.12) | 0.28 (0.24–0.32) |
| Eastern Europe | 692,424.46 (486,249.51–950,793.40) | 1,480.41 (1,130.73–1,879.5) | 1,044,207.33 (735,679.87–1,435,858.32) | 1,560.48 (1,197.51–1,981.88) | 0.1 (0.06–0.13) | 659,125.45 (314,711.85–1,351,113.76) | 1,409.22 (680.34–2,897.19) | 998,812.19 (478,498.91–2,036,260.71) | 1,492.64 (723.95–3,046.86) | 0.38 (0.32–0.44) |
| Western Europe | 1,643,336.13 (1,182,092.48–2,213,055.88) | 1,474.11 (1,132.83–1,870.11) | 2,694,205.44 (1,932,544.96–3,631,298.89) | 1,497.12 (1,151.23–1,902.91) | -0.08 (-0.12 to -0.04) | 1,495,084.70 (720,365.09–3,048,128.61) | 1,341.12 (655.82–2,731.06) | 2,555,759.31 (1,229,272.09–5,196,405.98) | 1,420.19 (691.09–2,882.20) | 0.17 (0.15–0.2) |
| Central Europe | 363,719.20 (255,657.03–499,242.02) | 1,384.02 (1,054.1–1,762.59) | 644,989.94 (452,469.04–887,982.08) | 1,444.55 (1,102.24–18,39.6) | 0.1 (0.08–0.12) | 309,077.73 (148,311.83–632,107.71) | 1,176.1 (571.26–2,408.74) | 590,665.68 (283,326.43–1,208,000.04) | 1,322.88 (643.24–2,708.41) | 0.45 (0.42–0.48) |
| High-income North America | 1,012,329.36 (712,676.10–1,383,940.68) | 1,473.13 (1,114.02–1,886.46) | 2,015,572.75 (1,423,276.63–2,754,319.17) | 1,570.91 (1,194.1–2,009.54) | 0 (-0.14 to 0.15) | 1,048,860.15 (506,753.24–2,134,104.26) | 1,526.28 (746.27–3,112.97) | 2,035,584.29 (987,053.55–4,123,622.18) | 1,586.51 (780.17–3,216.24) | -0.09 (-0.23 to 0.05) |
| Andean Latin America | 46,945.84 (33,680.48–63,485.76) | 1,466.03 (1,107.82–1,884.17) | 155,420.71 (111,370.97–209,999.52) | 1,548.7 (1,181.26–1,975.87) | 0.15 (0.14–0.17) | 39,330.92 (18,792.87–80,644.18) | 1,228.23 (583.82–2,544.71) | 137,623.27 (66,135.03–280,495.54) | 1,371.35 (659.82–2,812.61) | 0.37 (0.36–0.39) |
| Central Latin America | 192,093.74 (137,656.50–259,689.63) | 1,496.65 (1,139.6–1,910.11) | 662,864.09 (473,352.36–898,007.69) | 1,561.59 (1,190.42–1,988.76) | 0.12 (0.11–0.14) | 156,755.18 (75,353.29–319,274.14) | 1,221.31 (590.23–2,496.86) | 588,740.63 (283,179.21–1,199,360.37) | 1,386.97 (671.85–2,830.52) | 0.44 (0.43–0.45) |
| Caribbean | 65,435.26 (46,918.99–88,690.32) | 1,448.58 (1097.31–1,863.23) | 142,484.27 (101,779.30–193,155.19) | 1,498.33 (1,139.42–1,916.3) | 0.14 (0.13–0.16) | 54,987.77 (26,334.43–112,555.27) | 1,217.3 (582.89–2,516.83) | 127,424.33 (61,218.09–260,190.89) | 1,339.97 (643.47–2,748.65) | 0.34 (0.33–0.35) |
| Tropical Latin America | 221,453.50 (157,844.73–300,423.35) | 1,557.16 (1179.47–1,986.48) | 727,876.67 (516,874.33–993,234.35) | 1,635.98 (1,248.26–2,095.72) | 0.16 (0.16–0.17) | 173,092.63 (83,105.48–353,647.98) | 1,217.11 (585.98–2,497.19) | 619,901.85 (298,145.24–1,260,864.78) | 1,393.29 (676.54–2,843.37) | 0.46 (0.44–0.48) |
| Southern Latin America | 118,524.77 (85,082.50–160,315.47) | 1,445.71 (1,100.8–1,859.12) | 248,831.24 (177,331.71–336,106.00) | 1,534.2 (1,164.55–1,959.61) | 0.06 (0.03–0.1) | 109,246.70 (52,188.63–221,620.21) | 1,332.54 (636.75–2,727.42) | 238,388.93 (114,198.02–484,740.29) | 1,469.81 (708.73–2,999.40) | 0.3 (0.27–0.34) |
| Eastern Sub-Saharan Africa | 137,746.56 (98,566.00–186,613.71) | 1,273.85 (970.34–1,627.01) | 326,012.05 (232,400.11–443,048.73) | 1,376.68 (1,050.36–1,751.32) | 0.26 (0.25–0.27) | 98,299.32 (46,878.43–199,999.95) | 909.05 (435.02–1,856.78) | 251,621.21 (120,545.27–512,077.58) | 1,062.54 (511.88–2,167.50) | 0.54 (0.53–0.56) |

*(Continued)*

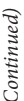

**Table 1.** (Continued)

| Characteristics | Incidence | | | | | DALYs | | | | |
|---|---|---|---|---|---|---|---|---|---|---|
| | Number of cases, 1990 | ASR per 100,000 population, 1990 | Number of cases, 2021 | ASR per 100,000 population, 2021 | EAPC, 1990–2021 | Number of cases, 1990 | ASR per 100,000 population, 1990 | Number of cases, 2021 | ASR per 100,000 population, 2021 | EAPC, 1990–2021 |
| Southern Sub-Saharan Africa | 61,648.71 (43,578.13–84,483.10) | 1457.02 (1,100.6–1,875.01) | 139,878.52 (99,202.29–191,232.05) | 1,557.82 (1,185.95–1,993.19) | 0.21 (0.2–0.22) | 51,494.26 (24,656.12–105,028.97) | 1,217.03 (582.21–2,502.68) | 118,629.32 (57,138.37–241,484.03) | 1,321.17 (640.39–2,700.41) | 0.31 (0.28–0.33) |
| Western Sub-Saharan Africa | 175,324.40 (124,822.61–238,569.75) | 1,308.35 (993.19–1,675.19) | 378,297.79 (269,166.16–514,907.99) | 1,383.63 (1,053.52–1,768.02) | 0.17 (0.16–0.19) | 131,856.49 (63,171.64–267,516.57) | 983.98 (472.12–2,000.09) | 40,024.02 (19,518.13–81,311.45) | 1,094.13 (528.82–2,230.35) | 0.37 (0.35–0.39) |
| North Africa and Middle East | 313,318.58 (223,554.50–424,856.92) | 1,266.6 (963.18–1,622.78) | 940,672.84 (672,709.08–1,275,126.73) | 1,376.75 (1,047.65–1,753.55) | 0.24 (0.22–0.25) | 233,987.36 (111,646.98–477,167.31) | 945.9 (453.71–1,934.24) | 99,450.96 (48,616.68–203,282.1) | 1,107.16 (537.57–2,266.12) | 0.52 (0.49–0.56) |
| Central Sub-Saharan Africa | 39,868.06 (28,427.95–54,024.95) | 1,311.3 (987.77–1,684.16) | 94,911.28 (67,561.96–128,418.92) | 1,359.4 (1,031.99–1,734.3) | 0.08 (0.05–0.11) | 29,842.37 (14,152.24–60,855.88) | 981.55 (461.79–2,020.09) | 11,066.43 (5,350.26–22,512.12) | 1,060.54 (506.79–2,171.93) | 0.23 (0.19–0.28) |

Abbreviations: ASR, age-standardised rate; CI, confidence interval; DALY, disability-adjusted life year; EAPC, estimated annual percentage change; GBD, global burden of disease; UI, uncertainty interval.

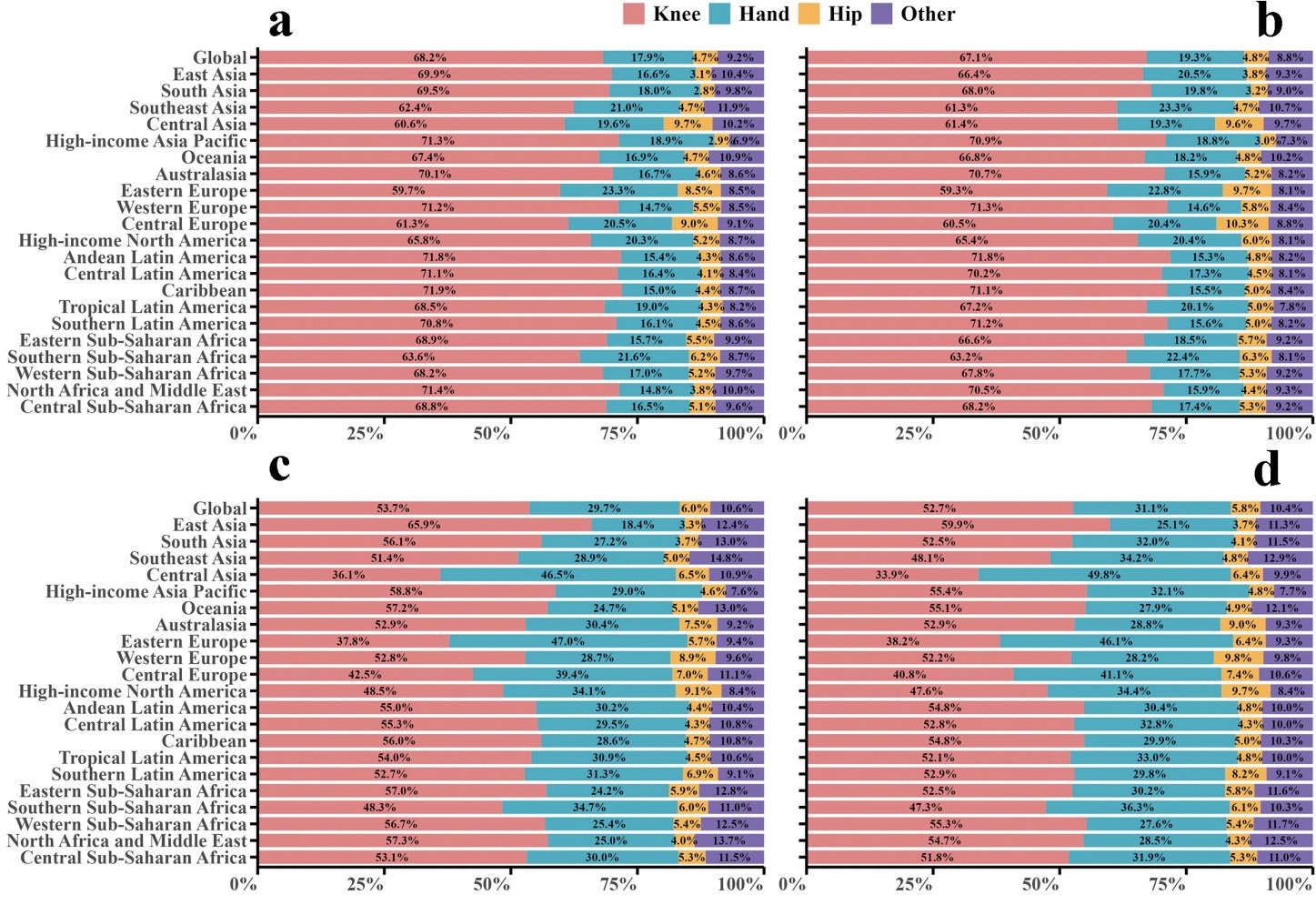

**Fig 1. Proportion of total incident cases and DALYs for osteoarthritis of different body parts among older adults globally and in 21 GBD regions in 1990 and 2021.** (a, b) Proportion of incidence cases in 1990 and 2021; (c, d) Proportion of DALYs in 1990 and 2021. DALY, disability-adjusted life-years.

the highest ASDR, at 1,766.13 per 100,000 (S2 Table in S1 File). Between 1990 and 2021, the ASR of OA in older adults showed an increasing trend in most countries. Equatorial Guinea exhibited the most significant growth, with an EAPC of 0.52 (95% CI: 0.48–0.56) for the ASIR and 1.29 (95% CI: 1.21–1.37) for the ASDR (Fig 2b, 2d).

Between 1990 and 2021, the ASR of OA in the four anatomical sites in older adults showed an increasing trend in most countries (S3–6 Tables and S2–5 Figs in S1 File). Among the 204 countries and territories, Grenada and the Maldives experienced the fastest increase in the ASIR and ASDR of knee OA, respectively. Ethiopia and Equatorial Guinea showed the fastest growth for hand OA. Equatorial Guinea exhibited the fastest growth for hip OA.

### Age and sex patterns in older adults

Globally, among older adults in 1990 and 2021, the peak incidence and number of DALYs of OA in the four anatomical sites occurred in the 65–69-year age group and gradually declined with increasing age. However, the trend for overall OA differed. The incidence rate declined with age, whereas the DALY rate increased with age, reaching its peak in the

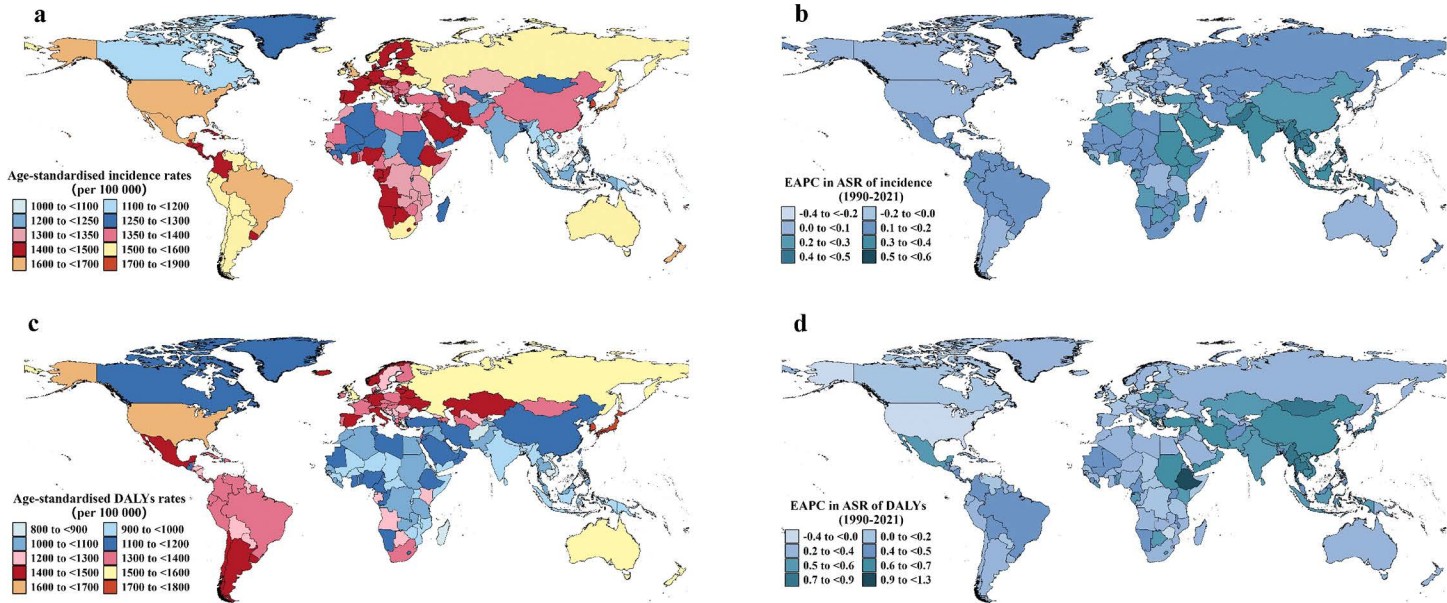

**Fig 2. Age-standardized incidence and DALY rates in 2021 and estimated annual percentage change from 1990 to 2021 for osteoarthritis among older adults across 204 countries and territories.** Age-standardized incidence (a) and DALY (c) rates and estimated annual percentage changes of age-standardized incidence (b) and DALY (d) rates. ASR, age-standardized rate; DALY, disability-adjusted life year; EAPC, estimated annual percentage change.

population aged ≥95 years. In addition, the incidence and DALY rates of female individuals were significantly higher than those of male individuals (Fig 3).

Between 1990 and 2021, the fastest global increase in incidence rate was observed for hip OA in the 80–84-year age group (EAPC: 0.48), whereas the fastest increase in DALY rate was noted for hand OA in the 65–69-year age group (EAPC: 0.49). Among the 21 GBD study regions, the most significant change in incidence rate was for hip OA in East Asia, peaking in the 65–69-year age group (EAPC: 1.15) (S6 Fig in S1 File).

## Association between ASR and SDI

From 1990 to 2021, the ASIR and ASDR of OA among older adults increased with increasing SDI both globally and across the study regions. The ASDR of OA in all four anatomical sites showed an increasing trend with increasing SDI. The ASIR for hip OA exhibited greater fluctuations (S7 Fig in S1 File).

In 2021, among 204 countries and territories, the ASIR and ASDR of OA in older adults showed a significant positive correlation with the SDI (r = 0.48, $P < 0.001$; r = 0.76, $P < 0.001$). A similar trend was observed for OA in all four anatomical sites (S8–12 Figs in S1 File).

## DALYs of OA attributable to high BMI in older adults

In 2021, 19.22% of the global ASDR of OA among older adults was attributable to high BMI, with PAFs of 35.09% for hip OA and 32.6% for knee OA. Notably, the ASDR attributable to high BMI was substantially higher for knee OA (220.11 per 100,000) than for hip OA (26.27 per 100,000) (S7 Table and S13 Fig in S1 File).

Age-stratified analysis revealed different patterns of PAFs across age groups. Among global older adults, total OA PAF showed a consistent decline from 20.61% in the 65–69-year age group to 14.96% in those aged 95 + years. Hip OA PAF decreased from 36.60% in the 65–69-year age group to 32.44% in the 80–84-year age group, and slightly increased to

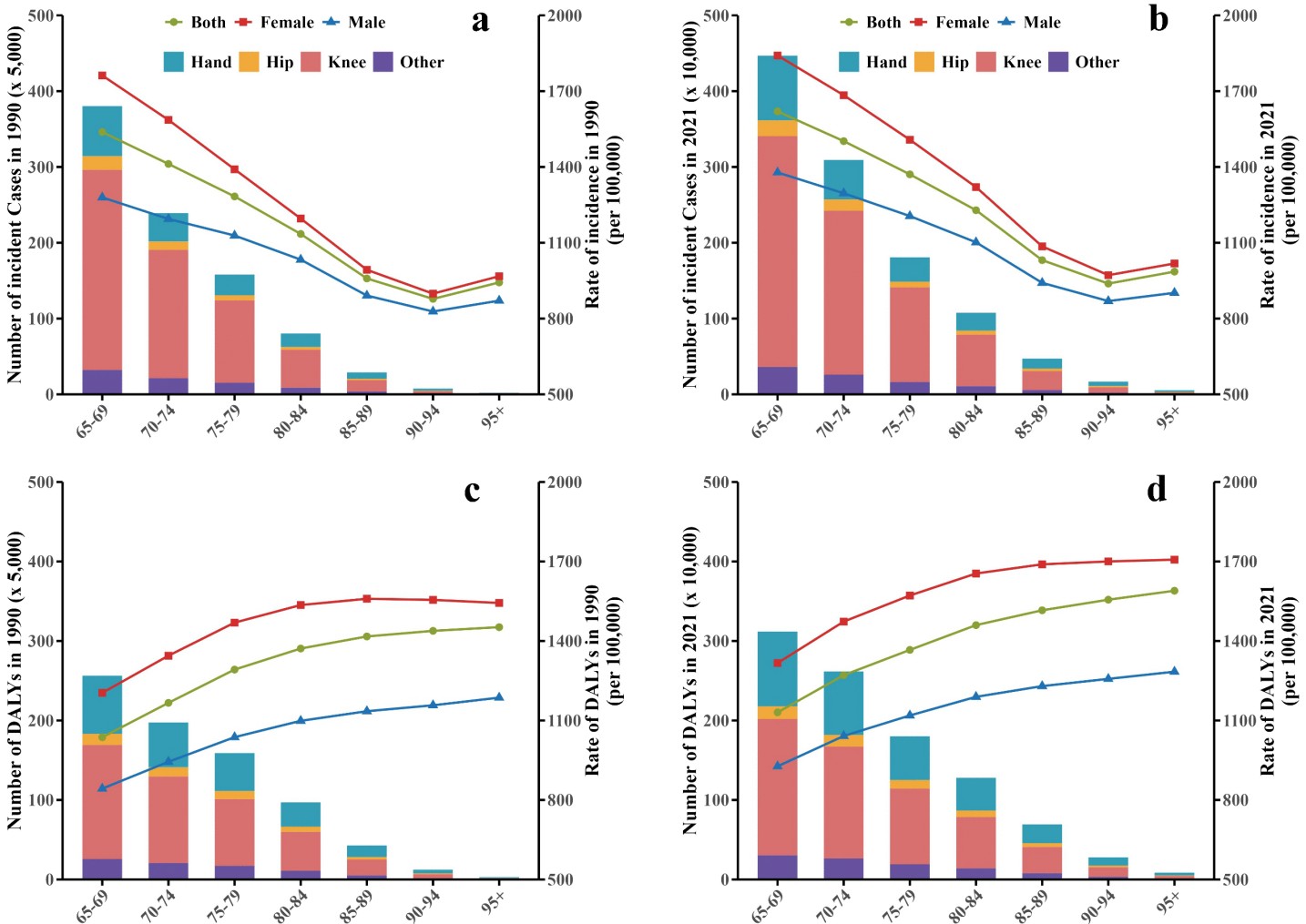

**Fig 3. Global incidence and DALYs of osteoarthritis in 1990 and 2021, overall and among older adults, by four anatomical sites, sex, and age group.** (a, b) Incidence cases and rates for 1990 and 2021; (c, d) DALYs and DALY rates for 1990 and 2021. Solid lines represent the age-standardized rates for osteoarthritis among adults aged 65 years and older. The four anatomical sites of osteoarthritis are the hand, hip, knee, and other sites. OA, osteoarthritis; DALY, disability-adjusted life-year.

31.20% in the 95 + age group. Knee OA PAF declined from 34.08% in the 65–69-year age group to its lowest point of 29.64% in the 80–84-year age group, subsequently rising to 31.16% in those aged 95 + years (Fig 4).

Among the 21 GBD study regions, Australasia had the highest overall OA PAF (26.81%), whereas hip and knee OA had the highest PAF in high-income North America (44.59% and 45.18%, respectively) (S7 Table in S1 File). Over the past 32 years, the global ASDR of OA attributable to high BMI among older adults increased from 186.73 to 246.38 per 100,000, with PAF rising by 21.39% since 1990. South Asia showed the fastest increase, with OA PAF increasing from 6.06% to 10.67%. At the national level, Ghana had the largest increase in OA PAF, increasing from 9.37% to 17.72% (S8 Table and S14 Fig in S1 File).

### Predictions of OA in older adults from 2020 to 2040

We used the BAPC model to predict the global disease burden trends of older patients with OA after 2021. The results indicate that during the period from 2021 to 2040, the ASIR of older patients with OA is expected to increase by 5.52%,

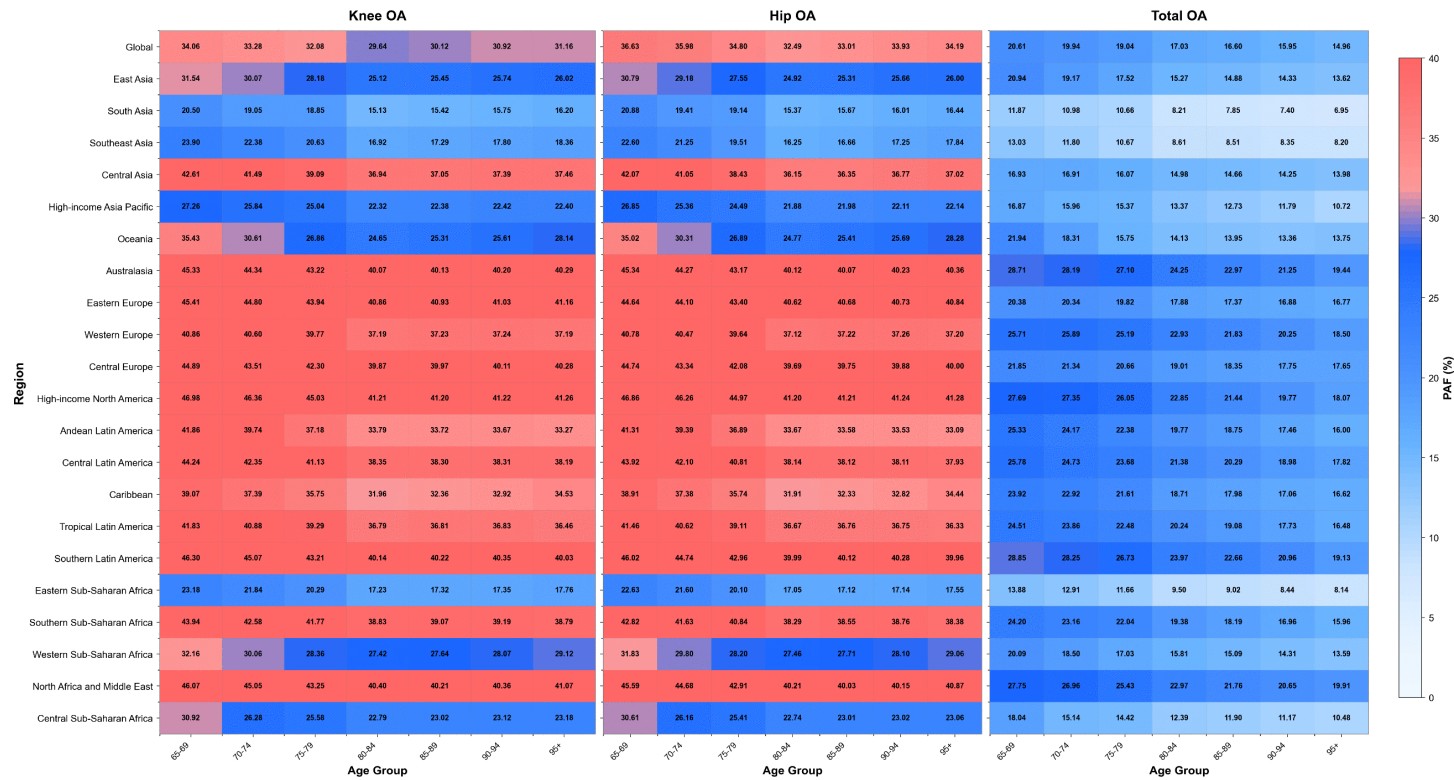

**Fig 4. Population attributable fractions for knee, hip, and total osteoarthritis attributable to high BMI among older adults across global and 21 regional populations, 2021.**

rising from 1446.09 per 100,000 to 1525.81 per 100,000. Correspondingly, the ASDR is projected to increase by 5.79%, from 1281.99 per 100,000 to 1356.20 per 100,000 (S9 Table in S1 File). Regarding sex, both the ASIR and ASDR are expected to remain consistently higher for females than for males (Fig 5). Regarding anatomical sites, knee OA in older adults will continue to dominate the burden from 2021 to 2040 (S9 Table in S1 File).

## Discussion

This study used estimates generated by the 2021 GBD modeling framework to describe long-term trends in the incidence and DALYs of OA among older adults. Because GBD estimates were derived from statistical models rather than direct observations, the reported patterns should be interpreted as modeled estimates. Within this context, we found that from 1990 to 2021, the incidence rate and DALYs for older adults increased globally. Knee OA imposed the heaviest burden, followed by hand OA. The incidence and DALY rates of OA across all four anatomical sites peaked in the 65–69-year age group, with a significantly higher burden observed among female individuals than among male individuals. The burden of overall OA and OA in different anatomical sites showed regional variations.

Previous studies have analyzed the burden of OA among adults aged 30–54 years or reported prevalence and incidence rates across all age groups globally and in different countries [14,16,30]. Unlike previous research, we comprehensively analyzed the overall, localized, and multidimensional trends in the burden of OA among older adults from 1990 to 2021, addressing the research gap for this age group and anatomical sites.

We observed that the global ASDR was lower than the ASIR for older adults. This pattern may reflect differences in age composition and disability levels within this group. However, these interpretations should be viewed tentatively, because

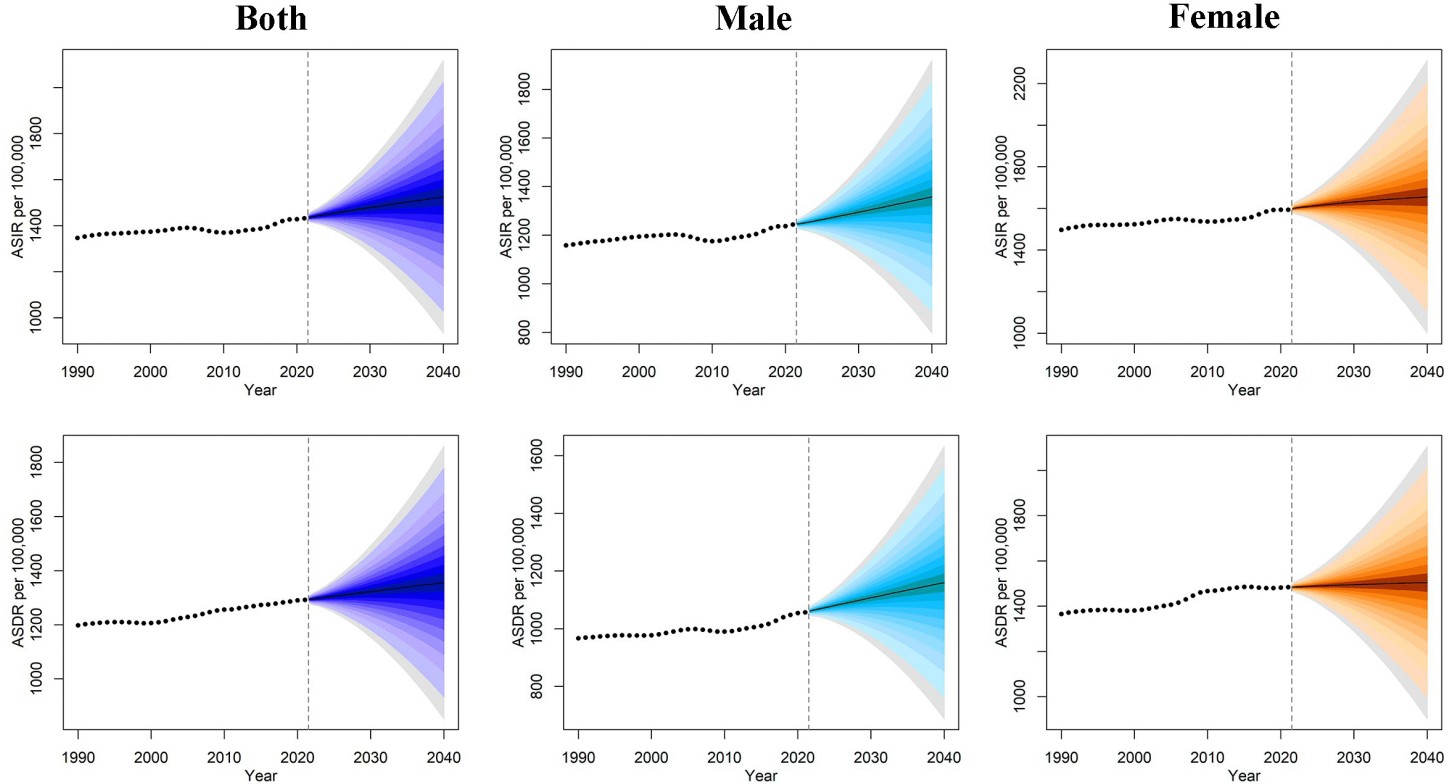

**Fig 5. Projected ASIR and ASDR for osteoarthritis among older adults at the global level by sex from 1990 to 2040 based on the BAPC model.**
ASIR: age-standardized incidence rate; ASDR: age-standardized DALY rate; BAPC, Bayesian Age-Period-Cohort.

the GBD modeling process cannot identify the exact mechanisms underlying variations between incidence and disability. The long-term trends were evaluated using the EAPC, which summarizes the average yearly change in ASRs based on a log-linear regression model. EAPC reflects model-derived trends and may be influenced by data completeness and modeling assumptions. Consistent with previous findings [5,14,21], the OA burden among older adults increased globally between 1990 and 2021. In 2020, OA was among the top 10 causes of YLD for adults aged ≥70 years, affecting one-third of individuals in this age group [6]. The prevalence of physician-diagnosed OA in the hand, knee, and hip increases with age, peaking in individuals in their 70s [21]. This indicates a growing disease burden of OA in the older adult population. Considering that OA incurs not only economic costs, such as medication, surgery, absenteeism, and premature death, but also individual costs, including pain, fatigue, and mobility restrictions, the medical costs of OA are estimated to account for 1.0–2.5% of the gross domestic product in many high-income countries, likely underestimating the actual burden [31].

In 2021, the OA burden among older adults was the highest in high-income Asia-Pacific and China, whereas Equatorial Guinea showed the fastest growth from 1990 to 2021. However, these differences should be interpreted with caution. Data availability varies widely across regions, and most input data come from high-income settings. Consequently, regional estimates may partly reflect modeling uncertainty rather than true epidemiological differences. Although factors such as health-system capacity or socioeconomic conditions may contribute to observed patterns, GBD data alone cannot confirm these mechanisms. Between 1990 and 2021, the burden of OA increased most rapidly in Southeast Asia. This may be due to the exacerbated vicious cycle of disease and poverty in low- and middle-income countries that often face issues such as uneven economic development, insufficient healthcare resources, and weak public health interventions, making

it difficult to effectively reduce the OA burden [32]. Additionally, the rapid ageing of the population in Southeast Asia may have contributed to the growing OA burden among older adults. It is projected that during 2025–2050, the number of the oldest adults (aged ≥80 years) will grow faster in Southeast Asia than in East Asia [33]. However, because GBD estimates cannot verify causal pathways, these potential explanations should be considered speculative rather than definitive.

The long-term trends of OA burden among older adults vary by anatomical site. Globally, the OA burden at the four anatomical sites is increasing, with knee OA having the greatest burden, followed by hand OA, whereas hip OA has the smallest burden, consistent with previous findings [14,16]. Specifically, early GBD analyses also identified knee OA as a major contributor to the global burden of osteoarthritis, focusing only on early-onset osteoarthritis or osteoarthritis in working-age populations [34,35]. However, compared with earlier studies, our analysis of adults aged ≥65 years reveals distinct age-specific patterns, particularly in the relative contributions and temporal trends across anatomical sites. For example, differences in trend magnitude between knee and hip OA appear more pronounced in older age groups. These differences may reflect age-related changes in joint vulnerability, survival with disability, and accumulated exposure to risk factors, as well as improvements in case detection and modeling in more recent GBD iterations, rather than true discrepancies in disease occurrence. By focusing on older adults, our study complements prior work and highlights aspects of OA burden that were previously underemphasized. OA was the fifth most costly disease treated in hospitals in the USA in 2008 [25]. A Dutch study estimated the economic burden of knee OA at €871 per person per month, factoring in both productivity losses and healthcare costs [36]. OA-related hospitalizations are often linked to joint replacement surgeries. In the USA, approximately 50% of joint replacement surgeries are knee replacements, 97% of which are performed to treat knee OA [25]. By 2030, the demand for primary total knee replacement surgeries is projected to increase to 3.48 million cases [37]. Given the substantial economic burden and the rapidly increasing treatment demand for knee OA in older adults, greater attention should be directed toward knee OA.

At the regional level, the burden of knee and hand OA increased most significantly in Southeast Asia, which is partly related to population ageing and partly to a potential link of OA with lower education levels and lower-skilled jobs [38]. This relationship can be explained by the fact that individuals with lower education levels are more likely to engage in prolonged physical labor [39]. At the national level, China, India, and the USA had the highest OA burden among older adults, which may be related to China and India have the largest populations globally, whereas the USA is the most populous developed country, leading to an excessive OA burden in these nations [20].

Similar to previous findings [40], we observed that the burden of OA peaked in the 65–69-year age group. Ageing is the most significant risk factor for OA, and the relationship between age and OA risk is likely multifactorial, involving factors such as oxidative damage, cartilage thinning, muscle weakness, and diminished proprioception [41]. Additionally, the OA burden was significantly higher in female individuals than in male individuals, a phenomenon also observed in previous studies [14,20]. The heavier burden among women may be associated with hormonal changes, although evidence remains inconsistent. Studies have suggested that the significant decline in estrogen after menopause may directly contribute to cartilage damage, bone loss, or increased susceptibility to OA in women. Furthermore, women often experience weight gain during menarche, menopause, and pregnancy [42]. However, the association between estrogen and OA has not been consistently confirmed in clinical and epidemiological studies [43].

Overall, the ASDR of OA among older adults showed a positive correlation with the SDI of GBD regions and countries. Socioeconomic status is closely associated with healthcare accessibility, with individuals of higher socioeconomic status having greater access to specialized medical services [44,45]. High-SDI regions, such as the high-income Asia-Pacific region, South Korea, and the USA, have more established social welfare systems, whereas lower-SDI regions are experiencing population ageing [46]. In high-SDI countries, more sedentary lifestyles and longer lifespans are common, which may contribute to a higher burden of OA [47]. Although factors such as access to medical services or lifestyle may influence the burden of OA, these explanations go beyond what can be inferred from the ecological estimates of the GBD and therefore should be regarded as hypotheses rather than confirmed mechanisms.

High BMI has been consistently associated with knee osteoarthritis in observational studies, and the GBD 2021 comparative risk assessment framework estimates that excess body weight contributes to a proportion of the global OA burden. However, these estimates are derived from a counterfactual modeling approach and should not be interpreted as evidence of causality. In 2021, 19.22% of the global age-standardized DALY rate of OA among older adults was attributable to high BMI, with higher population-attributable fractions observed for hip OA (35.09%) and knee OA (32.60%) than for total OA.In addition to the overall attributable burden, age-stratified patterns suggest heterogeneity in BMI-attributable OA burden across older age groups, indicating that the contribution of high BMI is not uniform throughout later life. In this study, BMI-attributable estimates were not stratified by sex. Previous studies have reported sex differences in the association between BMI and OA burden in older populations, potentially reflecting biological, behavioral, or social factors [29,48]; however, such differences could not be evaluated within the current analytical framework.Given that high BMI is currently the only OA risk factor with a quantified attributable burden in the GBD study, our findings should be interpreted as population-level descriptive estimates rather than as indicators of individual-level risk or intervention effectiveness. Future research that quantifies the attributable burden of other modifiable risk factors, such as occupational exposures and joint injury, would provide a more comprehensive understanding of OA etiology and burden in aging populations.

Our study has some limitations. First, all GBD estimates are produced through a modeling framework rather than direct measurement. Therefore, the accuracy of incidence and DALY estimates depends heavily on the completeness and quality of input data, which are uneven across regions and particularly limited in many low- and middle-income countries. These gaps may introduce uncertainty and contribute to underestimation or overestimation of the true burden. Second, heterogeneity in diagnostic criteria and reporting practices across sources may lead to misclassification, constraining the comparability of estimates across countries and time. Third, the attribution of OA to high BMI is derived from a theoretical counterfactual model and should not be interpreted as causal. BMI was not included as a covariate for hand OA owing to insufficient evidence, limiting comparability across anatomical sites. Moreover, the inclusion of spinal osteoarthritis within the low back and neck pain category may lead to underestimation of site-specific OA. Finally, important modifiable factors such as occupational exposures and joint injuries are not yet incorporated into the GBD risk framework, which may result in an incomplete assessment of the attributable burden. These limitations underscore the need for more population-based registries, harmonized diagnostic criteria, and longitudinal studies to validate and refine modeled estimates.

## Conclusions

We comprehensively analyzed the incidence and DALY rates of OA in older adults globally from 1990 to 2021 and projected the global trend of OA in older adults up to 2040 using the BAPC model, providing valuable insights for policymakers, healthcare providers, and researchers. The burden of OA among older adults varies significantly across regions and anatomical sites, driven primarily by population ageing in recent years. In ageing societies, healthcare providers should optimize region-specific and OA-type-specific medical interventions to address this challenge more effectively. The study findings highlight the significant impact of high BMI on the OA burden among older adults and projects that the burden of OA will continue to rise in the future. Public awareness of modifiable risk factors for OA, such as weight management and healthy lifestyles, should be increased to help reduce the overall OA burden.

## Supporting information

**S1 File. OA and BMI trends in older adults globally.**
(DOCX)

## Acknowledgments

We appreciate the works by the Global Burden of Diseases, Injuries, and Risk Factors Study (GBD) 2021 collaborators.

## Author contributions

**Conceptualization:** Yan Gao, Wenfeng Han.

**Data curation:** Yan Gao, Hailong Yu, Ning Wang.

**Formal analysis:** Yan Gao, Aoxiang Yang.

**Funding acquisition:** Yu Wang.

**Methodology:** Yan Gao, Hailong Yu, Ning Wang, Bin Zheng, Aoxiang Yang.

**Software:** Yan Gao.

**Supervision:** Yu Wang.

**Visualization:** Yan Gao, Wenfeng Han, Ning Wang, Bin Zheng.

**Writing – original draft:** Yan Gao.

**Writing – review & editing:** Yan Gao, Wenfeng Han.

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
