## [Decision Letter · Decision Letter 0]

23 Nov 2025

Dear Dr. Wang,

Thank you for submitting your manuscript to PLOS ONE. After careful consideration, we feel that it has merit but does not fully meet PLOS ONE’s publication criteria as it currently stands. Therefore, we invite you to submit a revised version of the manuscript that addresses the points raised during the review process.

According to the reviewers' comments, the authors need to clarify methodology, results, and discussion. Also, need to explain the study limitations.

We look forward to receiving your revised manuscript.

Kind regards,

Victoria Pando-Robles, Ph.D.

Academic Editor

PLOS ONE

Journal Requirements:

“This work was supported by the Liaoning Provincial Science and Technology Plan Joint Program (Applied Basic Research Project) [grant number 2023JH2/101700124].”

3. PLOS requires an ORCID iD for the corresponding author in Editorial Manager on papers submitted after December 6th, 2016. Please ensure that you have an ORCID iD and that it is validated in Editorial Manager. To do this, go to ‘Update my Information’ (in the upper left-hand corner of the main menu), and click on the Fetch/Validate link next to the ORCID field. This will take you to the ORCID site and allow you to create a new iD or authenticate a pre-existing iD in Editorial Manager

4. Please include captions for your Supporting Information files at the end of your manuscript, and update any in-text citations to match accordingly. Please see our Supporting Information guidelines for more information: http://journals.plos.org/plosone/s/supporting-information ..

5. We note that Figure 2 in your submission contain [map/satellite] images which may be copyrighted. All PLOS content is published under the Creative Commons Attribution License (CC BY 4.0), which means that the manuscript, images, and Supporting Information files will be freely available online, and any third party is permitted to access, download, copy, distribute, and use these materials in any way, even commercially, with proper attribution. For these reasons, we cannot publish previously copyrighted maps or satellite images created using proprietary data, such as Google software (Google Maps, Street View, and Earth). For more information, see our copyright guidelines: http://journals.plos.org/plosone/s/licenses-and-copyright.

Reviewers' comments:

Reviewer's Responses to Questions

**Comments to the Author**

1. Is the manuscript technically sound, and do the data support the conclusions?

Reviewer #1: Partly

Reviewer #2: Yes

2. Has the statistical analysis been performed appropriately and rigorously?

Reviewer #1: No

Reviewer #2: I Don't Know

3. Have the authors made all data underlying the findings in their manuscript fully available?

Reviewer #1: Yes

Reviewer #2: Yes

4. Is the manuscript presented in an intelligible fashion and written in standard English?

Reviewer #1: Yes

Reviewer #2: No

Reviewer #1: The manuscript addresses an important and timely public health issue: the global burden of osteoarthritis in older adults. The use of GBD 2021 provides access to a comprehensive and standardized epidemiological dataset. The focus on adults ≥65 years is clinically relevant, and the inclusion of site-specific OA estimates (hip, knee, and hand), high BMI–attributable burden, and future projections adds potentially useful detail for health planning. The manuscript is generally well written and the descriptive results are clearly presented.

The major limitations relate to methodological transparency and reproducibility. The manuscript relies heavily on secondary GBD data but does not sufficiently detail how metrics were accessed, processed, or analyzed. The Bayesian age–period–cohort modeling used for projections is insufficiently described, lacking information on model specifications, prior distributions, diagnostics, and sensitivity analyses. As written, the projections cannot be evaluated or replicated. Some interpretations overreach the limits of ecological, modeled GBD data—for example, implying causal links between BMI and OA or discussing estimates as if they reflect directly observed incidence. The discussion would benefit from a more thorough acknowledgment of the inherent limitations of GBD modeling and the risk-factor attribution framework. Overall, the conceptual contribution remains limited, and several findings reiterate trends previously reported in GBD publications.

The study offers only modest originality. While GBD-based burden analyses of OA are common, this manuscript adds some incremental value by focusing specifically on older adults, incorporating site-specific burden estimates, and exploring high BMI–attributable DALYs within this age group. However, these contributions are relatively minor and do not represent substantial methodological or conceptual innovation. This level of originality is acceptable for PLOS ONE provided the methods are fully transparent and the conclusions remain appropriately constrained.

The manuscript provides an incremental update to current understanding of OA burden, primarily through: age-focused analysis of adults ≥65 years, description of site-specific patterns, quantification of BMI-attributable burden in older adults, and projections of future trends. These insights are useful for health system planning and reinforce well-established global trends. However, the contribution is limited by the methodological weaknesses noted above, which must be addressed for the findings to be interpretable and reliable.

Reviewer #2: Dear Authors,

Thank you for submitting your manuscript. Your study highlights a globally important public-health issue. The manuscript is generally clear and well-structured, but several points need clarification and improvement:

1. Introduction

• Clearly state study objectives in 2–3 numbered points and highlight novelty (e.g., global burden in older adults, BMI attribution, projection of trends).

• Justify the chosen time period for the analysis and the projection horizon. Explain why these years were selected and why projections are made to this particular future year.

2. Methods

• Data sources: Specify countries/regions covered, data type (survey, registry, claims), and years included.

• Modeling approach: Clarify which model was used (e.g., DisMod-MR version, Bayesian APC methods), priors, software, and convergence diagnostics.

• Uncertainty intervals: Explain how they were computed.

• OA joint locations: Clarify which joints were included in BMI-attributable fraction calculations (e.g., knee, hip, hand).

3. Results and Discussion

• Trends over time: Explain clearly how OA prevalence trends were calculated. Are the reported increases due to real epidemiological changes or data differences across years?

• BMI-OA association: The manuscript mentions that higher BMI is linked to OA, but quantitative details (effect sizes, stratification by age/sex) are limited. Including specific numbers would strengthen this point.

• Comparison with previous studies: Comparisons are mostly narrative. More detailed discussion of how your findings align or differ from prior research, with possible explanations, would enhance the discussion.

• Limitations: Some limitations are noted, but key issues regarding data quality across countries, temporal coverage, and differentiation of BMI effects by OA location are not fully addressed. Expanding on these would provide a more balanced view.

Addressing these points will clarify your study’s contribution and improve its impact.

**Do you want your identity to be public for this peer review?** For information about this choice, including consent withdrawal, please see our For information about this choice, including consent withdrawal, please see our Privacy Policy .

Reviewer #1: No

Reviewer #2: No

---

## [Author Response · Author response to Decision Letter 1]

18 Dec 2025

Dear Dr. Pando-Robles,

Thank you very much for giving us the opportunity to revise our manuscript entitled "Osteoarthritis in older adults: a global health challenge and the role of high BMI in shaping disease trends" (Manuscript ID: PONE-D-25-34533).

We sincerely appreciate the time and effort you and the reviewers have dedicated to providing insightful comments and constructive suggestions. We have carefully reviewed all the comments and have made substantial revisions to the methodology, results, and discussion sections to address the concerns raised. We believe these revisions have significantly improved the quality and rigor of our manuscript.

Below, we provide a point-by-point response to the Journal Requirements and the Reviewers’ comments. All changes have been highlighted in the "Revised Manuscript with Track Changes" file.

Part I: Response to Journal Requirements

Comment:

Response:

We have carefully reviewed the PLOS ONE style guidelines and reformatted the manuscript, title page, and file names accordingly.

2. Financial Disclosure

Comment:

Please clarify the sources of funding, the role of funders, and salary information.

Response:

We have updated our financial disclosure as requested. The detailed statement is included in the cover letter and below:

Funding Sources:

This work was supported by the Liaoning Provincial Science and Technology Plan Joint Program (Applied Basic Research Project) (2023JH2/101700124) and the Liaoning Provincial Science and Technology Plan Joint Program (Technology Research and Development Project) (2024JH2/102600270).

Role of Funders:

Salary:

None of the authors received a salary or personal remuneration from these funding sources.

3.Supporting Information Captions

Comment:

Please include captions for your Supporting Information files at the end of your manuscript.

Response:

We have added a list of captions for all Supporting Information files at the end of the manuscript and updated the in-text citations.

4.Figure 2 Copyright

Comment:

We note that Figure 2 in your submission contain [map/satellite] images which may be copyrighted... We require you to either (1) present written permission... or (2) remove the figures... or (3) supply a replacement figure.

Response:

We confirm that Figure 2 was created entirely by the authors using R software (version 4.2.3) with the ggplot2 and sf packages. The map boundaries were derived from public domain shapefiles (e.g., Natural Earth data), and no proprietary data sources (such as Google Maps or satellite imagery) were used. Therefore, the authors hold the copyright, and the figure is fully compatible with the PLOS ONE CC BY 4.0 license.

Part II: Response to Reviewer #1

Point 1: Methodology Transparency

Comment:

The manuscript relies heavily on secondary GBD data but does not sufficiently detail how metrics were accessed, processed, or analyzed.

Response:

We have substantially expanded the Methods section to clarify how GBD 2021 data were accessed, processed, and analyzed (Page 6, lines 122–132). Specifically, we now state that all estimates were extracted from the GBD 2021 results tool, with annual data on incidence and DALYs for osteoarthritis and its four anatomical sites (knee, hip, hand, and other OA) retrieved for the period 1990–2021. We have clarified the geographic coverage (204 countries and territories, aggregated into 21 GBD regions), stratification by sex and seven older age groups (65–69 to ≥95 years), and the use of 95% uncertainty intervals derived from the 1000-sample posterior distributions as provided by the GBD 2021 study. These additions improve the transparency and reproducibility of the data extraction and analytical procedures.

Point 2: BAPC Modeling Details

Comment:

The Bayesian age–period–cohort modeling used for projections is insufficiently described (model specifications, priors, diagnostics, sensitivity analyses).

Response:

We have revised the Methods section to provide a clearer description of the projection approach (Page 8, lines 169–178). We now explicitly state that a BAPC model was used, with age, period, and cohort effects modeled using second-order random walk priors and posterior inference conducted via Integrated Nested Laplace Approximation (INLA). We have clarified that all model specifications followed the default settings of the “BAPC” R package and that IHME population projections were incorporated to account for future demographic changes. We also emphasize that the projections are descriptive in nature and intended to characterize long-term trends rather than to infer causal relationships. These additions improve methodological transparency and allow readers to better assess the assumptions underlying the projections.

Point 3: Interpretation of Ecological Data (Causality)

Comment:

Some interpretations overreach the limits of ecological, modeled GBD data—for example, implying causal links between BMI and OA.

Response:

Specifically, we revised the sections related to high BMI to avoid causal language and to clarify that BMI-attributable estimates are derived from a counterfactual modeling approach rather than directly observed data. We now explicitly state that these estimates should not be interpreted as evidence of causality and represent population-level attributable fractions rather than individual-level risk (Page 30, Lines 424–441). In addition, we revised the Results and Discussion to caution against interpreting modeled estimates as directly observed incidence or disability. We now note that differences between the ASIR and ASDR should be interpreted tentatively, as the GBD modeling framework cannot identify the mechanisms underlying these variations (Page 26, Lines 340–343; Page 27, Lines 356–362).

Point 4: Study Limitations

Comment:

The discussion would benefit from a more thorough acknowledgment of the inherent limitations of GBD modeling.

Response:

We have expanded the Limitations section to more explicitly acknowledge the constraints of GBD-based analyses (Page 30, Lines 442–457). We now clarify that GBD estimates are derived from statistical modeling rather than direct measurement, are subject to data quality and coverage limitations across regions, and may be affected by heterogeneity in diagnostic and reporting practices. We also explicitly state that BMI-attributable estimates are based on a counterfactual model and should not be interpreted as causal, note limitations in site-specific attribution, and acknowledge that several important modifiable risk factors are not yet included in the GBD risk framework.

Part III: Response to Reviewer #2

Point 1: Introduction (Objectives & Justification)

Comment:

Clearly state study objectives in 2–3 numbered points... Justify the chosen time period and projection horizon.

Response:

We revised the latter part of the Introduction to more clearly articulate the study rationale, objectives, and justification of the study period and projection horizon (Page 4, Lines 62–79). We now explicitly highlight the limited number of studies focusing on older adults using the latest GBD 2021 data, and clarify the knowledge gaps regarding BMI-attributable burden and future projections in this population. We further justify the selection of 1990–2021 as the study period to capture the most comprehensive long-term historical trends available in GBD 2021, and explain that projections to 2040 were chosen to provide a ~20-year horizon relevant to population ageing and long-term public health planning. Finally, we explicitly state the study objectives in three numbered points, covering burden estimation and trends, BMI-attributable burden, and future projections using the BAPC model.

Point 2: Methods (Data Sources & Modeling)

Comment:

Clarify data sources, modeling approach, uncertainty intervals, OA joint locations included in BMI-attributable estimates, and projection methods..

Response:

We revised the Methods section to provide a clearer and more detailed description of data sources, modeling procedures, uncertainty estimation, risk attribution, and projections.

Specifically, we now describe the scope and composition of the GBD 2021 data, including the number and types of data sources, geographic coverage (204 countries and territories across 21 GBD regions), and study period (1990–2021), as well as the use of publicly available datasets curated through the GHDx (Page 4, Lines 83–104). We clarify that incidence estimates were generated using the DisMod-MR 2.1 Bayesian meta-regression framework and that DALYs were calculated as the sum of YLLs and YLDs following standard GBD methods.

We also explicitly describe the use of 95% uncertainty intervals derived from the 1000-sample posterior distributions provided by GBD 2021 (Page 6, Lines 128–131). Regarding risk attribution, we clarify that high BMI is the only OA risk factor quantified in GBD 2021 and that BMI-attributable burden was estimated using the population attributable fraction framework for total OA, knee OA, and hip OA, with age-stratified analyses conducted at global and regional levels (Page 8, Lines 161–168).

Finally, we expanded the description of the projection approach, specifying the use of a Bayesian age–period–cohort model with second-order random walk priors, posterior inference via INLA, incorporation of IHME population projections, and the descriptive (non-causal) nature of the projections (Page 8, Lines 169–178).

Point 3: Results (Trends Calculation)

Comment:

Explain clearly how OA prevalence trends were calculated. Are the increases due to real changes or data differences?

Response:

We clarified in the Methods section that long-term trends in osteoarthritis burden were evaluated using the estimated annual percentage change (EAPC), which summarizes the average yearly change in age-standardized rates based on a log-linear regression model (Page 8, Lines 151–155). A positive EAPC with a 95% confidence interval excluding zero indicates a significant increasing trend, while a negative EAPC indicates a decreasing trend.In the Discussion, we further clarify that EAPC reflects model-derived trends in age-standardized rates and does not distinguish between true epidemiological changes and variation arising from data availability, quality, or modeling assumptions (Page 26, Lines 340–346). We explicitly caution that observed differences between incidence and disability metrics, as well as long-term trends, should be interpreted tentatively, as the GBD modeling framework cannot identify the underlying mechanisms driving these patterns.

Point 4: BMI-OA Association Details

Comment:

Provide quantitative details (effect sizes, stratification by age/sex) for the link between BMI and OA.

Response:

We revised the Methods, Results, and Discussion sections to provide more explicit quantitative details on the association between high BMI and OA burden, consistent with the GBD 2021 comparative risk assessment framework.

In the Methods, we clarify that high BMI is the only OA risk factor quantified in GBD 2021 and that BMI-attributable burden was estimated using the population attributable fraction (PAF) approach for total OA, knee OA, and hip OA. We describe how BMI-attributable proportions of age-standardized DALY rates were calculated globally, across 21 regions and 204 countries and territories, and further stratified by age group at the global and regional levels (Page 8, Lines 161–168).

In the Results, we now present age-stratified PAF estimates, demonstrating heterogeneous patterns across older age groups. Specifically, global PAFs for total OA declined from 20.61% in adults aged 65–69 years to 14.96% in those aged ≥95 years, while hip and knee OA showed distinct age-related trajectories (Page 24, Lines 292–298; Fig 4).

In the Discussion, we further report that in 2021, 19.22% of the global age-standardized DALY rate of OA among older adults was attributable to high BMI, with higher attributable fractions for hip OA (35.09%) and knee OA (32.60%) than for total OA. We also explicitly state that BMI-attributable estimates were not stratified by sex in the present study and acknowledge that previously reported sex differences could not be evaluated within the current analytical framework (Page 30, Lines 424–441). We emphasize that these estimates are population-level, model-based, and should not be interpreted as evidence of individual-level causality or intervention effects.

Point 5: Comparison with Previous Studies

Comment:

Comparisons with previous studies are mostly narrative. A more detailed discussion of similarities and differences between your findings and prior studies, with possible explanations, would strengthen the discussion.

Response:

We revised the Discussion to strengthen comparisons with previous GBD studies and to move beyond a purely narrative description (Page 27, Lines 372–385). We now explicitly distinguish findings that are consistent with earlier analyses—such as knee OA remaining the largest contributor to global OA burden—from differences observed in adults aged ≥65 years, including more pronounced age-specific variation in site-specific burden and temporal trends. We further provide plausible explanations for these differences, including age-related changes in joint vulnerability, survival with disability, cumulative risk exposure, and improvements in data coverage and modeling in recent GBD iterations.

Point 6: Study Limitations

Comment:

Some limitations are noted, but key issues regarding data quality across countries, temporal coverage, and differentiation of BMI effects by OA location are not fully addressed. Expanding on these would provide a more balanced view.

Response:

We expanded the Limitations section to more clearly and comprehensively address the constraints of this study (Page 30, Lines 442–457). We now explicitly acknowledge that GBD estimates are derived from a modeling framework rather than direct measurement and are subject to uncertainty arising from uneven data availability and quality across regions, especially in low- and middle-income countries. We also note potential misclassification due to heterogeneity in diagnostic criteria and reporting practices. In addition, we clarify that BMI-attributable estimates are based on a counterfactual model and should not be interpreted as causal, that BMI was not included for hand OA due to insufficient evidence, and that site-specific OA burden may be underestimated because spinal OA is classified under low back and neck pain. Finally, we acknowledge that several important modifiable risk factors are not yet included in the GBD risk framework, which may result in an incomplete assessment of attributable burden.

---

## [Decision Letter · Decision Letter 1]

19 Feb 2026

Osteoarthritis in older adults: a global health challenge and the role of high BMI in shaping disease trends

PONE-D-25-34533R1

Dear Dr. Wang,

We’re pleased to inform you that your manuscript has been judged scientifically suitable for publication and will be formally accepted for publication once it meets all outstanding technical requirements.

Kind regards,

Victoria Pando-Robles, Ph.D.

Academic Editor

PLOS One

Additional Editor Comments (optional):

Reviewers' comments:

Reviewer's Responses to Questions

**Comments to the Author**

Reviewer #1: All comments have been addressed

2. Is the manuscript technically sound, and do the data support the conclusions?

Reviewer #1: Yes

3. Has the statistical analysis been performed appropriately and rigorously?

Reviewer #1: Yes

4. Have the authors made all data underlying the findings in their manuscript fully available?

Reviewer #1: Yes

5. Is the manuscript presented in an intelligible fashion and written in standard English?

Reviewer #1: Yes

Reviewer #1: (No Response)

**Do you want your identity to be public for this peer review?** For information about this choice, including consent withdrawal, please see our For information about this choice, including consent withdrawal, please see our Privacy Policy .

Reviewer #1: **Yes:** Daniel Xibillé-FriedmanDaniel Xibillé-Friedman

---

## [Editor Report · Acceptance letter]

PONE-D-25-34533R1

PLOS One

Dear Dr. Wang,

I'm pleased to inform you that your manuscript has been deemed suitable for publication in PLOS One. Congratulations! Your manuscript is now being handed over to our production team.

Kind regards,

on behalf of

Victoria Pando-Robles

Academic Editor

PLOS One